# Construction of disease-specific cytokine profiles by associating disease genes with immune responses

**Tianyun Liu**[1], **Shiyin Wang**[2], **Michael Wornow**[3], **Russ B. Altman**[4]*

1 Department of Bioengineering, Stanford University, Stanford, California, United States of America, 2 Chinese Undergraduate Visiting Research Program, Stanford University, Stanford, California, United States of America, 3 Department of Computer Science, Stanford University, Stanford, California, United States of America, 4 Departments of Bioengineering, Genetics & Medicine, Stanford University, Stanford, California, United States of America

* russ.altman@stanford.edu

**Data Availability Statement:** All relevant data are available at https://github.com/TianyunC/cytokine-networks.

**Funding:** RBA received support from NIH GM102365. The funders had no role in study

## Abstract

The pathogenesis of many inflammatory diseases is a coordinated process involving metabolic dysfunctions and immune response—usually modulated by the production of cytokines and associated inflammatory molecules. In this work, we seek to understand how genes involved in pathogenesis which are often not associated with the immune system in an obvious way communicate with the immune system. We have embedded a network of human protein-protein interactions (PPI) from the STRING database with 14,707 human genes using feature learning that captures high confidence edges. We have found that our predicted Association Scores derived from the features extracted from STRING's high confidence edges are useful for predicting novel connections between genes, thus enabling the construction of a full map of predicted associations for all possible pairs between 14,707 human genes. In particular, we analyzed the pattern of associations for 126 cytokines and found that the six patterns of cytokine interaction with human genes are consistent with their functional classifications. To define the disease-specific roles of cytokines we have collected gene sets for 11,944 diseases from DisGeNET. We used these gene sets to predict disease-specific gene associations with cytokines by calculating the normalized average Association Scores between disease-associated gene sets and the 126 cytokines; this creates a unique profile of inflammatory genes (both known and predicted) for each disease. We validated our predicted cytokine associations by comparing them to known associations for 171 diseases. The predicted cytokine profiles correlate (p-value<0.0003) with the known ones in 95 diseases. We further characterized the profiles of each disease by calculating an "Inflammation Score" that summarizes different modes of immune responses. Finally, by analyzing subnetworks formed between disease-specific pathogenesis genes, hormones, receptors, and cytokines, we identified the key genes responsible for interactions between pathogenesis and inflammatory responses. These genes and the corresponding cytokines used by different immune disorders suggest unique targets for drug discovery.

design, data collection and analysis, decision to publish, or preparation of the manuscript.

**Competing interests:** The authors have declared that no competing interests exist.

## Author summary

The success of anti-TNF treatment in multiple inflammatory diseases suggest that there is a shared cytokine framework that defines highly conserved mechanisms of inflammation. However, clinical trials testing the efficacy of new cytokine inhibitors suggest a more complex set of interacting cytokine mechanisms that are associated with different diseases. In this work, we aim to define the disease-specific role of cytokines that mediate pathogenesis and inflammatory processes, focusing on autoimmune diseases. We hypothesize that specific clinical phenotypes result from the interactions between disease-specific cytokines and disease-related genes (identified through genetics, transcriptomics, and analysis of metabolic dysfunctions), even though they also may share a common cytokine elements and conserved mechanisms of inflammation. We have developed novel network methods that show a robust ability to identify differential associations between characteristic cytokines and genetics factors contributing to pathogenesis. We have validated our methods on 171 well-studied diseases; the predicted associations between cytokines and disease modules correlate with the published data. Our predictions provide the underlying difference of molecular mechanisms that may be responsible for clinical phenotypes.

## Introduction

The pathogenesis of inflammatory diseases is a coordinated process involving metabolic dysfunctions, signaling, and innate immune response—modulated by the production of cytokines and associated inflammatory molecules [1,2,3]. The continued discovery of novel pathways and inflammatory mediators reveals the complexity and richness of autoimmune diseases [4,5], but the complete molecular decision network behind these processes and the coordination between cytokine signaling and underlying disease biology are not well understood [6,7,8]. Many models of autoimmune diseases posit a common cytokine framework with highly conserved mechanisms of inflammation [4,9,10]. Recent advances in genome-wide association studies (GWAS) provide evidence of considerable genetic overlap between autoimmune diseases, along with unique loci for individual diseases—and sometimes their subtypes [11,12,13]. The success of anti-TNF treatment in multiple inflammatory diseases suggests that there is a shared cytokine framework (at least for a subset of diseases) that defines conserved mechanisms of inflammation [14]. However, clinical trials testing the efficacy of cytokine inhibitors suggest a more complex set of interacting cytokine mechanisms that are associated with different disease phenotypes [15]. For example, the Jak-Stat-Socs signaling module can have either pro- or anti-inflammatory outcomes depending on the activation pattern of cytokine receptors—and different diseases show different patterns. If diseases do not have the same cytokine activity profile, then it is important to define which ones are key for each disease. Therefore, we ask the question: What is the degree to which cytokine responses are shared across diseases or specific to each disease? And how does heterogeneity of cytokine responses mediate different pathogenesis and inflammatory processes?

Addressing the above questions requires a deeper understanding of the connections between cytokine signaling and the disease-specific genes implicated in pathogenesis of specific diseases and discovered through GWAS or transcriptional studies. Networks of interacting genes provide a useful representation of the functional associations between genes and gene modules [16,17]. Unfortunately, efforts to identify disease-associated genes do not always provide a clear link between pathogenesis and immune response. Publicly available immunology databases often limit their coverage to only characterizing properties of immune processes.

ImmuneSigDB maps changes in the expression of sets of genes corresponding to immune response, thereby enabling a system analysis of data to improve the understanding of immune processes [18]. ImmProt [19] uses high-resolution mass spectrometry-based proteomics to characterize primary human immune cell types for ligand and receptor interactions, thereby connecting distinct immune functions. ImmuneXpresso [20] identifies relationships between cells, cytokines, and diseases via Natural Language Processing (NLP) of PubMed articles.

Ideally, we would like to map known inflammatory response genes (e.g., cytokines and cytokine receptors) more completely to human gene networks to better identify potential mechanistic links between inflammatory response genes and pathogenesis genes. STRING is a comprehensive gene network that focuses chiefly on molecular pathways, with cancer heavily emphasized. Unfortunately, STRING does not provide fully elaborated links to immune response [21]. We have compared the network sparsity (the ratio of the number of links present in a graph to the total number of possible links that would be present in a complete graph) within and between known immune and disease-related functional modules in STRING (Fig A in S1 Text). We have found that the associations between different immune response genes (as identified by ImmProt) are under-represented in the STRING database. We hypothesize that either the connections are more difficult to study and characterize or that the connections are less dense compared with metabolic or signaling modules as the human body requires more traffic for metabolic or signaling activities.

Nevertheless, we aim to bridge this gap between pathogenesis and the immune processes that ultimately cause systemic damage. This will allow us to understand how immune responses are triggered in a disease-specific manner. We hypothesize that specific clinical phenotypes result from the interactions between disease-specific cytokines and disease-related genes (identified through genetics, transcriptomics, and analysis of metabolic dysfunctions), even though they also may share a common cytokine elements and conserved mechanisms of inflammation. In this study, we identify these disease-specific cytokines and their associated disease-specific genes to provide insights into the underlying molecular mechanisms. These mechanisms may suggest new approaches to treatment and treatment combinations for specific clinical phenotypes.

## Results

### A complete map of associations between 14K human genes

We developed a novel network method to construct a complete interaction map between human genes by embedding a PPI of 14,707 human genes using network scalable feature learning [22] that captures 728,090 high confidence edges in STRING [21] (Fig 1A and Table A in S1 Text). We have calculated Association Scores of all possible pairs (108,140,571 pairs) between the 14,707 human genes. The distribution of the predicted Association Scores of all possible pairs (108,140,571 pairs) is similar with that of the known pairs in STRING (9,250,034 pairs) (Figs B-D in S1 Text). Among these pairs, STRING provides a confidence index for 9,250,034 pairs, of which 8,521,944 have a confidence index below 800 (signifying high quality) and thus these lower quality pairs were not used for embedding. Figs 2 and 3 show that the predicted Association Scores correlate with the level of confidence in STRING (scatter plot shown in Fig B in S1 Text).

The 9,250,034 pairs are grouped into four boxplots based on their predicted Association Scores (Fig 2). As the predicted Association Score increases, the average STRING confidence index also increases. Association Scores below 0.6 have a low average STRING confidence of ~200. On the other hand, ~75% of Association Scores above 0.8 correspond with STRING confidence indexes above 800 (Fig 3). Thus, it appears that the predicted Association Scores

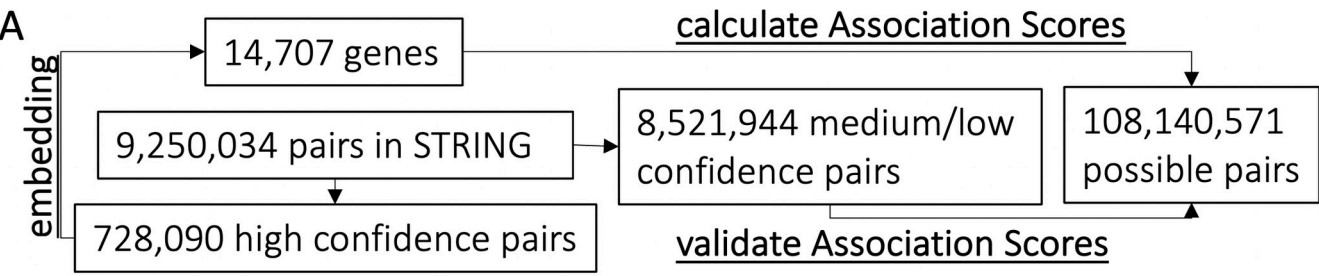

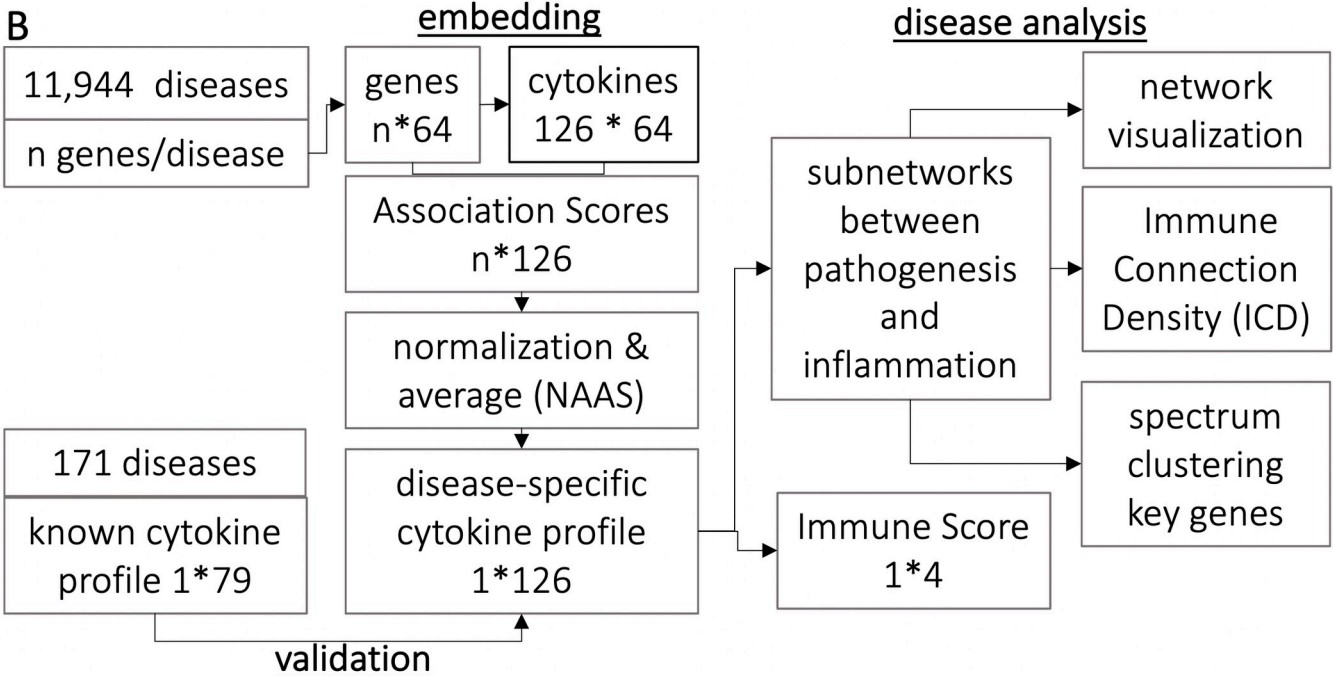

**Fig 1.** Data flow for (A) predicting Association Scores and (B) analyzing disease-specific cytokine profiles. A. Network features of the high confidence STRING pairs were used to embed 14,707 human genes. The predicted associations between the 14,707 genes were validated using medium and low confidence STRING pairs. B. For each gene associated with a given disease, we calculated Association Scores with each of the 126 cytokines. The Association Scores were averaged and normalized to NAAS that represent the cytokine profile of the given disease. These profiles were further analyzed by (1) calculating Immune Scores and (2) analyzing subnetworks formed between pathogenesis and inflammation by employing network visualization, spectrum partition, and estimation of connection density.

between genes in our embedded network are good predictors of protein-protein interactions, which can thereby enable construction of a complete and reliable network between all 14,707 human genes in STRING.

## Essential cytokines can be classified into six clusters based on their interaction profiles

We identified 126 well-studied cytokines based on ImmuneXpresso [20] in our embedded network. We call these 126 cytokines "essential cytokines" because they are linked to at least one disease in the literature. The term "cytokine" encompasses inflammatory regulators, including

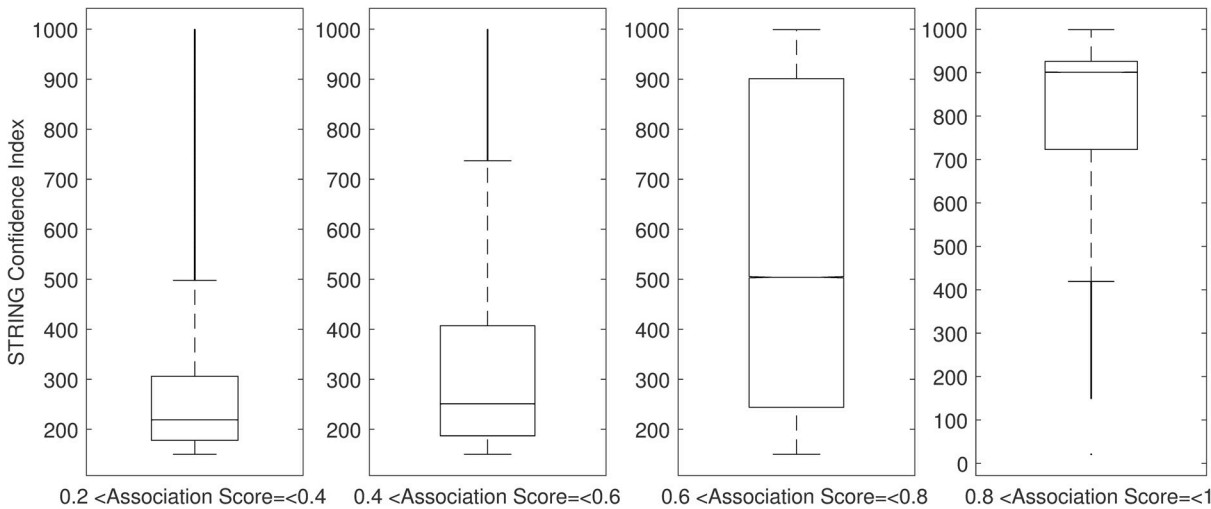

**Fig 2. The predicted Association Score between two genes measures the confidence of their associations.** We calculated Association Scores for all possible pairs (108,140,571 pairs) of 14,707 human genes. A total of 9,250,034 of these pairs have a known STRING confidence index. The confidence indexes of known STRING pairs are shown in the four boxplots below, grouped by their predicted Association Scores. As the predicted Association Score increases (left to right), the average STRING confidence index also increases (low to high).

interferons, the interleukins, the chemokine family, mesenchymal growth factors, the tumor necrosis factors, and other non-classified ones [23]. Each essential cytokine can be described by its location in the embedded network space, and its Association Scores with each of the 14,581 non-cytokine human genes in our network may suggest known or novel interactions with other human genes (Fig 1B). The Association Scores between each of the cytokines and the 14,581 non-cytokine human genes classified these 126 cytokines generally into six distinct

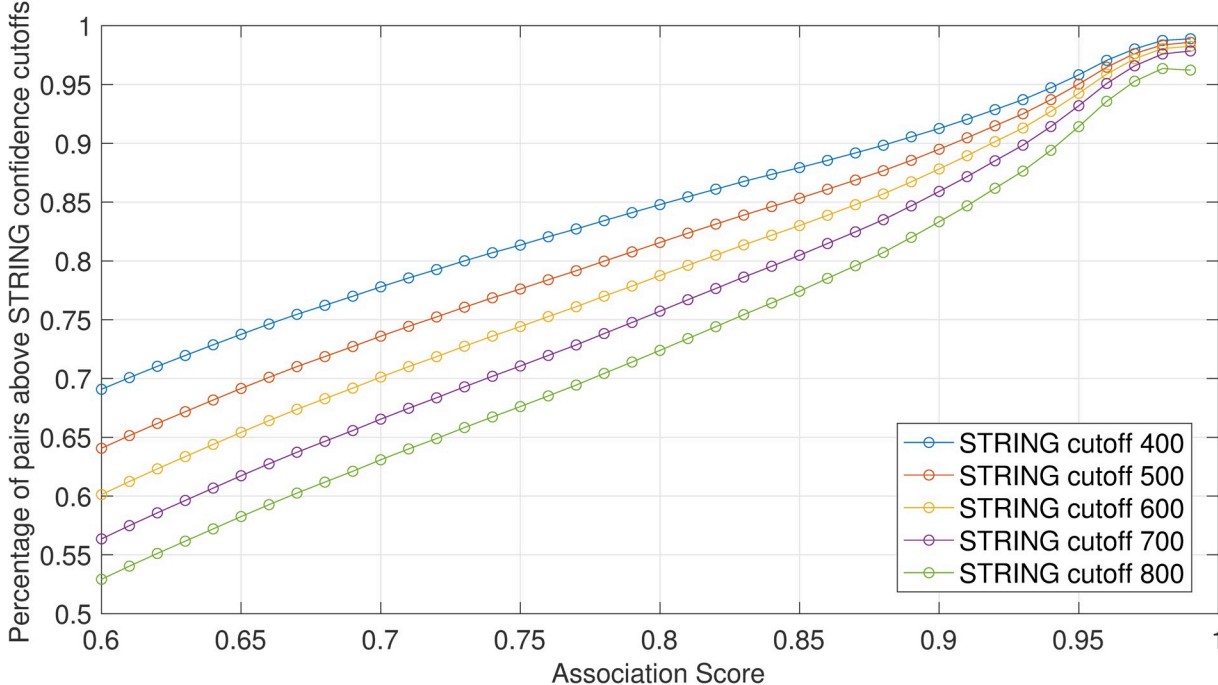

**Fig 3. The predicted Association Score between two genes measures the confidence of their associations.** The percentage of known STRING pairs above a certain confidence index cutoff. (Note: STRING confidence indexes are discrete scores).

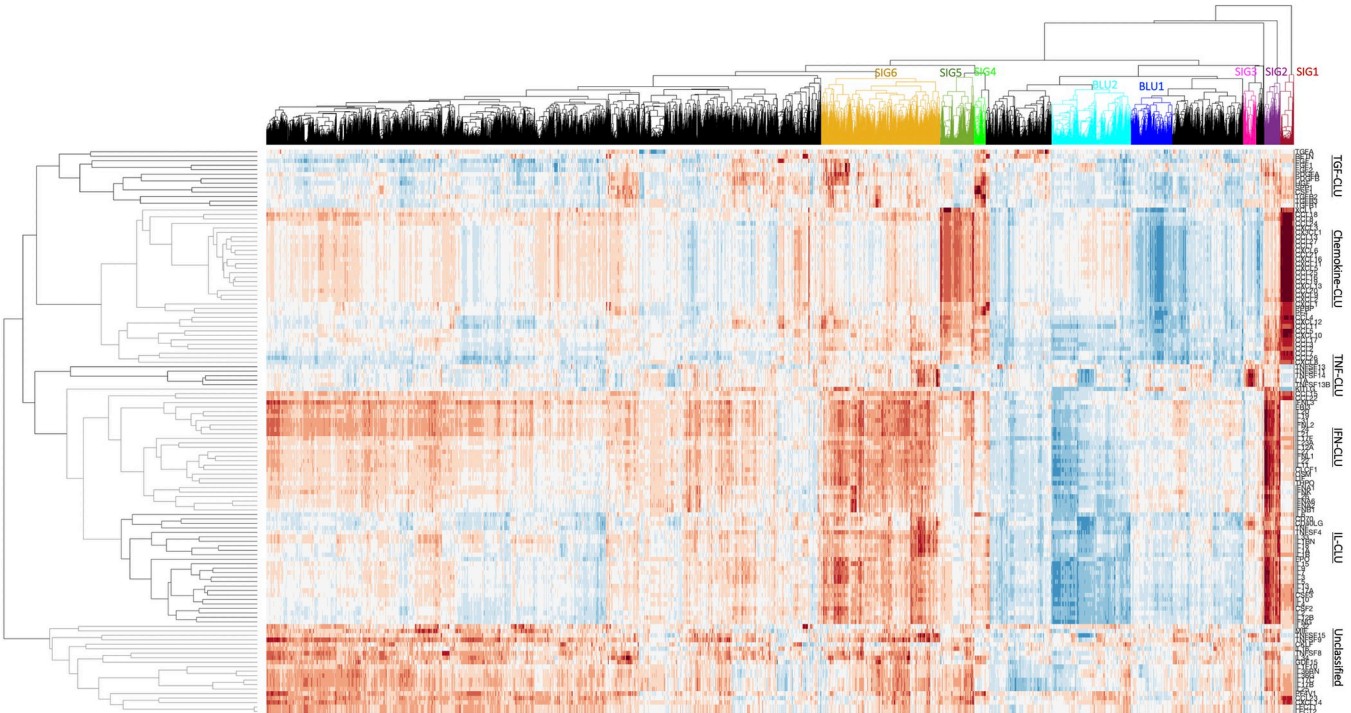

**Fig 4. The 126 cytokines form six clusters based on their Association Scores with the 14,581 non-cytokine genes.** The six clusters are named after their most enriched types of cytokines: TGF-CLU (growth factors), Chemokine-CLU (chemokines), TNF-CLU (TNFs), IFN-CLU (interferons), IL-CLU (interleukins), and Unclassified-CLU. Based on the dendrogram of the hierarchical cluster tree, we identified six gene sets (SIG1-SIG6) that associate with the six individual clusters and two gene sets (BLU1, BLU2) that do not interact with the three major clusters (Chemokine-CLU, IFU-CLU, IL-CLU). Details of cytokines and signatures are in Tables 1–4.

clusters, which we have named based on the types of their most enriched cytokine/chemokines: TGF-CLU (growth factors), Chemokine-CLU (chemokines), TNF-CLU (TNFs), IFN-CLU (interferons), IL-CLU (interleukins), and Unclassified-CLU (Fig 4). Unfortunately, it remains difficult to quantify the enrichment and purity of these clusters, due to the pleiotropy and element of redundancy in the cytokine family (each cytokine has many overlapping functions, and each function is mediated by multiple cytokines.) Six sets of close interactors are suggested by the dendrogram of the hierarchical cluster tree (names as SIG1-SIG6 in Table 1) and allow us to capture the function of each cluster with a gene signature. For example, Chemokine-CLU contains genes involved in the G-protein signaling pathway as well as cellular responses to endogenous and environmental insults, while TGF-CLU contains with genes that mediates blood coagulations and plasminogen activating cascades which are often associated with the innate immunity in infectious and neuroinflammatory diseases (Tables 2 and 3). We also identified two sets of genes

**Table 1. The associations between the six gene sets (SIG1-SIG6) and specific clusters (CLU).**

|  | TGF-CLU | Chemokine-CLU | TNF-CLU | IFN-CLU | IL-CLU | Unclassified-CLU |
|---|---|---|---|---|---|---|
| SIG1 |  | Yes |  |  |  |  |
| SIG2 |  |  |  | Yes | Yes |  |
| SIG3 |  |  | Yes |  |  |  |
| SIG4 | Yes |  |  |  |  |  |
| SIG5 |  | Yes |  |  |  |  |
| SIG6 |  |  |  | Yes | Yes |  |

**Table 2. Specific cytokines in each of the six clusters.**

| TGF-CLU | Chemokine-CLU | TNF-CLU | IFN-CLU | IL-CLU | Unclassified-CLU |
|---|---|---|---|---|---|
| TGFA RETN EGF FGF1 FGF2 PDGFA PDGFB HGF SPP1 CSF1 TGFB2 TGFB3 TGFB1 | XCL1 CCL18 CCL8 CCL24 CXCL3 CX3CL1 CCL13 CCL27 CCL1 CXCL6 CCL21 CXCL16 CXCL11 CXCL5 CCL25 CCL16 CCL19 CXCL13 CCL20 CXCL9 CXCL2 CXCL1 PPBP PF4 CCL4 CXCL12 CCL11 CCL5 CXCL10 CCL17 CCL3 CCL2 CCL7 CCL26 CXCL8 | TNFSF13 TNFSF11 TNFSF14 LTA TNFSF13B KITLG | CCL15 CCL22 IFNL3 EBI3 IL20 IL19 IL31 IFNL2 IL24 IL21 IL17F IL23A IL12A IL27 IFNL1 IL22 IL11 CLCF1 OSM LIF THPO IFNA1 IFNK IL26 IFNA6 IFNA2 IFNB1 | IL6 CD70 CD40LG TNF TNFSF4 IL33 IL1RN IL18 IL1A IL1B EPO IL15 IL9 IL7 IL3 IL5 IL13 IL17A CSF3 IL10 IL4 CSF2 IL2 IL12B IFNG | IL32 MIF TNFSF15 TNFSF9 CKLF IL16 TNFSF8 IL34 GDF15 IL1F10 IL36RN IL36G IL17C IL17B IL25 PF4V1 CCL23 CXCL14 LECT1 LECT2 |

(BLU1, BLU2) that are distant from the three major clusters (Chemokine-CLU, IFU-CLU, IL-CLU) (Table 4). Both BLU1 and BLU2 have biological functions in the nucleus. The full gene lists for each of the six clusters are listed in Table B in S1 Text.

## The predicted cytokine profiles of 171 diseases are validated using literature

We collected gene sets for 11,944 diseases from DisGeNET [24]. A majority (5,048) of these diseases are linked with fewer than ten genes, while a few are associated with up to 2,000 genes (Figs D-F in S1 Text).

For each disease, we estimated the normalized average Association Scores (NAAS) between each cytokine and the disease based on the normalized score of averaged Association Scores of the given gene set (Fig 1B), resulting in a 126-dimensional cytokine profile for each disease, which represents its disease-specific cytokine profile. From the overall set of 11,944 diseases, we identified 171 well-studied diseases whose co-occurrence frequencies with the cytokines (for 79 of our 126 essential cytokines) had been evaluated by ImmuneXpresso through literature sampling. The predicted profiles correlate with the literature sampling frequency significantly (p-value cutoff with multiple testing correction is 0.0003) for 95 of the 171 diseases, suggesting reasonable reliability of predicted cytokine profiles (Fig 5 and Table C in S1 Text). Note that when the number of disease-associated genes increases, the accuracy of the predicted cytokine profiles decreases (Fig G in S1 Text). Fig 6 shows an example of the predicted

**Table 3. The functional annotations of the six genes sets (SIG1-SIG6).**

| | GO/PANTHER Pathways |
|---|---|
| SIG1 | G-protein signaling pathways, inflammation mediated chemokine pathways, GABA-B receptor signaling, 5HT1 receptor signaling, Enkephalin release, Endothelin signaling, opioid pathways, dopamine pathways, Nicotine pharmacodynamics, Metabotropic glutamate pathways |
| SIG2 | Interleukin signaling, JAK/STAT signaling, Toll receptor signaling, Interferon-gamma signaling, inflammation mediated by cytokine signaling pathways |
| SIG3 | Ubiquitin proteasome pathways, Parkinson disease, Hedgehog signaling, WNT signaling, FGF signaling, Angiogenesis, hypoxia response via HIF activation, p53 pathways by glucose deprivation |
| SIG4 | Blood coagulation, Plasminogen activating cascade |
| SIG5 | G-protein signaling pathways, 5HT2 receptor signaling, Histamine H1 receptor signaling, Thyrotropin releasing hormone receptor signaling, Oxytocin receptor signaling, Muscarinic acetylcholine receptor signaling. Corticotrophin releasing factor receptor signaling, Angiotensin II stimulate signaling, Endogenous cannabinoid signaling |
| SIG6 | Apoptosis signaling, Toll receptor signaling, FAS signaling, p38 MAPK pathway, Gonadotropin-releasing hormone receptor pathways, B-cell activation, T-cell activation, Insulin/IGF pathway (protein kinase B cascade, MAPKK/MAPK cascade), axon guidance mediated by netrin, Angiogenesis, EGF signaling, PDGF signaling, VEGF signaling |

**Table 4. The two gene sets (BLU1, BLU2) that do not interact with the three major clusters Chemokine-CLU, IFU-CLU, and IL-CLU.**

|  | GO/PANTHER Pathways |
|---|---|
| BLU1 | Ribosome biogenesis, rRNA metabolic process, gene expression, translation |
| BLU2 | DNA repair, DNA metabolic process, telomere maintenance, cellular macromolecule metabolic process, response to stress |

cytokine profiles for aneurysm, a disease that is not typically considered as an immune disorder. The NAAS between aneurysm and each of the 79 cytokines of which the literature sampling frequency in diseases are known in ImmuneXpresso are plotted with known associations (frequency cutoff is 0.005) marked in solid blue squares. At a high cutoff (NAAS >0.8), the recall rate is 21/24. The novel predictions are: HGF, IL11, IL12B, IL13, IL15, IL17F, IL22, IL33IL5, IL7, IL9, LIF, OSM, PDGFA, PDGFB, PPBP. A scatter plot showing the correlation between the literature co-occurrence frequency and the predicted NAAS in aneurysm is in Fig H in S1 Text).

## Defining the key modes of cytokine response

We have shown that our predicted disease profiles for 79 cytokines align with known cytokine associations, strengthening the validity of our predicted profiles for all 126 cytokines which include novel predictions for 47 cytokines. There are many ways to classify diseases, we aimed to assess patterns in cytokine response for different diseases, reasoning that shared cytokine response might indicate the potential for shared treatment strategies. We analyzed the predicted disease profiles of 126 cytokines in the 171 well-studied diseases. The 171 diseases

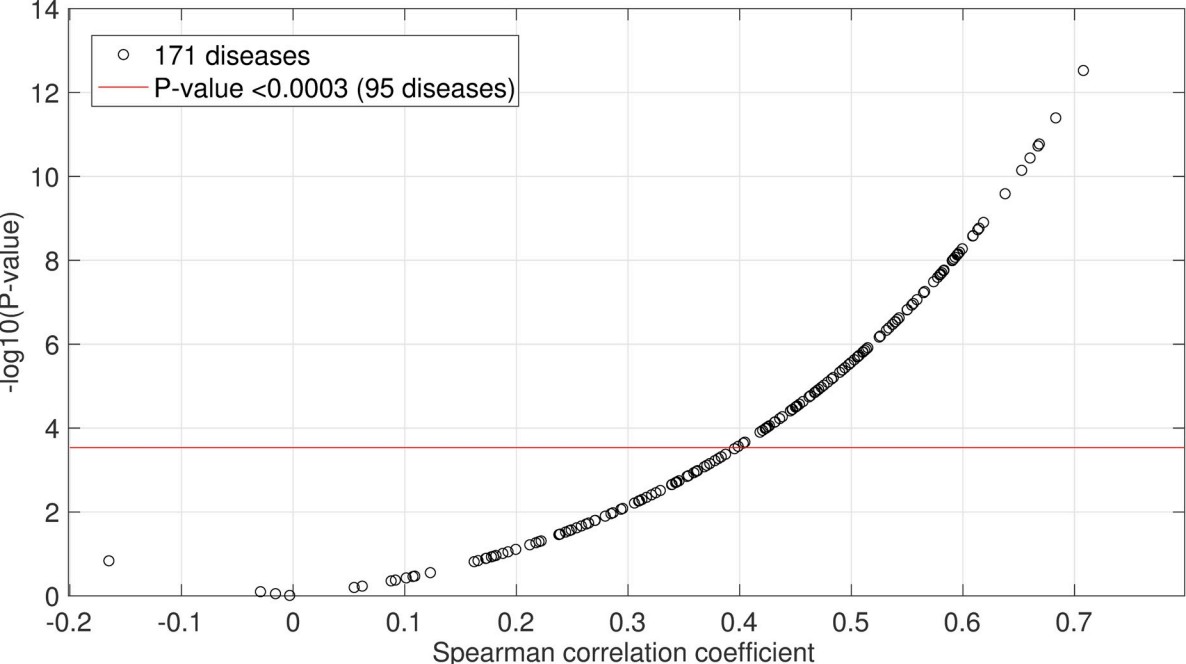

**Fig 5. Predicted cytokine profiles for 171 well-studied diseases correlate with cytokine sampling in literature.** The Spearman correlation coefficients between each disease's NAAS and known literature sampling frequency are plotted against the P-values. Of the 171 diseases, we were able to predict the profiles for 95 diseases with p-value<0.0003 (corrected cutoff by multiple testing), suggesting the accuracy of the predicted profiles.

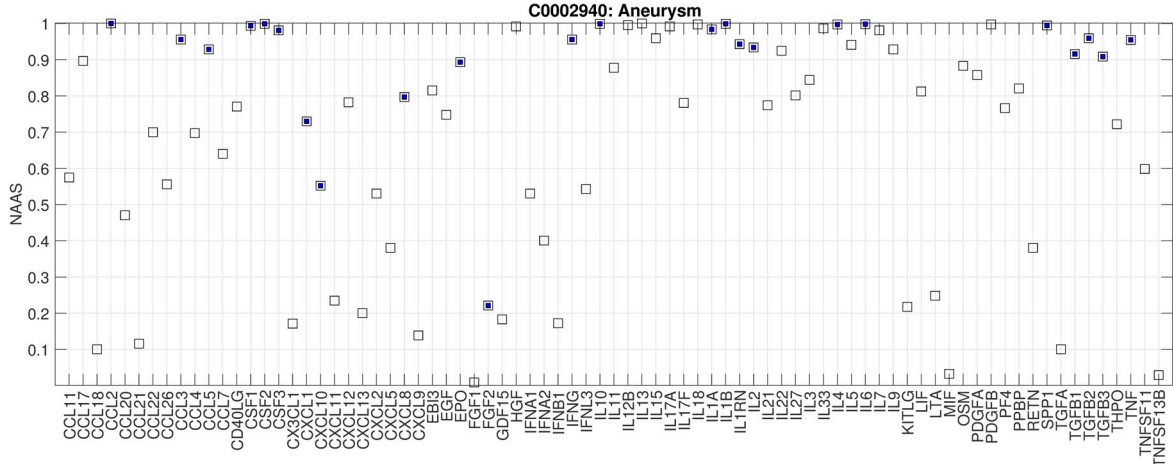

**Fig 6. The NAAS between aneurysm and each of the 79 cytokines for which the literature sampling frequency in disease is known in ImmuneXpresso.** Known associations (frequency cutoff of 0.005 in ImmuneXpresso) are marked in solid blue squares.

include 23 immune disorders (C20), 48 infections (C01), seventeen cardiovascular diseases (C14), thirteen metabolic disorders (C18), and 55 neoplasms (C04). Note that one disease may belong to more than one disease classes (Table D in S1 Text). These diseases separate into three primary clusters based on their interactions with cytokines (Fig 7). As shown in Fig 7, immune disorders are enriched in cluster labeled "cluster-1". Infections are split into two clusters ("cluster-1" and "cluster-2"). Note that of the twenty neoplasms in cluster-1, nineteen are hematologic or lymphatic diseases (C15/C04), suggesting that these neoplasms have distinct cytokine distributions from other neoplasms. Most metabolic diseases (11/13) and cardiovascular disorders (11/17) are enriched in cluster-3 (blue dendrogram in Fig 7), suggesting that cytokine responses are shared across different disease classes. The three clusters of cytokine responses that form across these five classes of disease suggest that there are common cytokine frameworks shared across disease classes, even though there is also significant heterogeneity of cytokine response within each disease class.

The disease cytokine profiles show that inflammatory cytokines that include interleukins, inferons, TNFs are grouped together (Fig 7), while chemokines form two groups based on their associations with diseases. These groups of cytokines drive the clustering of diseases across different classes. Therefore, we analyzed the influence on diseases from inflammatory components, chemokines, growth factors, and other cytokines. We defined Immune Scores to capture the contributions from four categories of cytokines to the inflammation process of a given pathogenesis (Fig 8). In general, immune disorders and infections show higher values for all four cytokine-type components, with cardiovascular diseases presenting intermediate values, and metabolic diseases and, specially, neoplasm, having the lowest average immune scores for all four components. The chemokine scores of immune disorders spread in a wide range. Infections have the highest scores for growth factors. Cardiovascular diseases have higher scores than metabolic diseases in all four categories, while neoplasms show the lowest scores in all the categories. In summary, a disease can be represented by a 126-dimension cytokine profile or a 4-dimensional summary Immune Scores, both of which suggest that the inflammatory responses in different diseases are mediated by different distributions of cytokines.

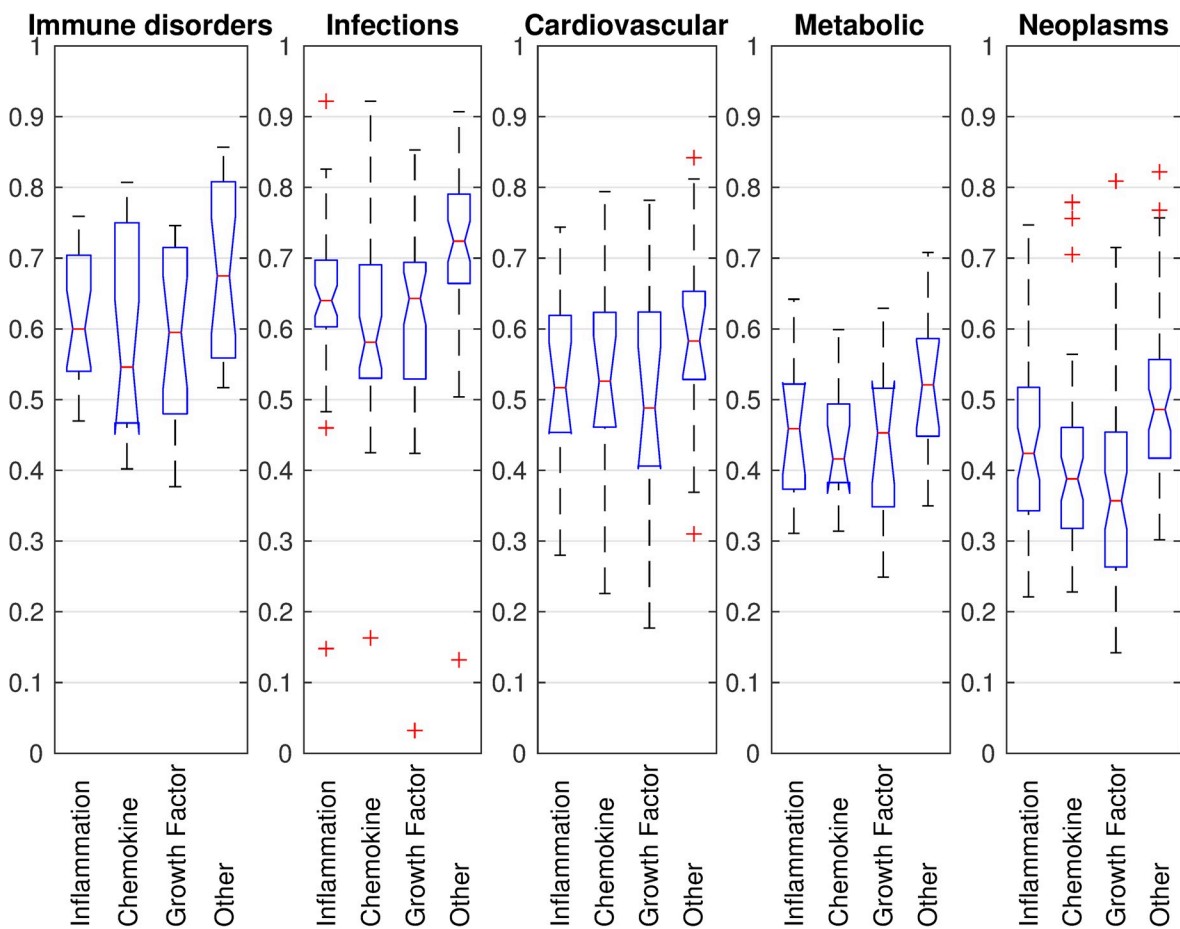

**Fig 7. Cytokine features for the 171 well-studied diseases.** The 171 diseases formed three clusters based on their NASS with different types of cytokines. Immune disorders are enriched in cluster-1. Infections are split into two clusters (cluster-1 and cluster-2). Note that of the twenty neoplasms in cluster-1, nineteen are hematic and lymphatic diseases (C15/C04). Most metabolic diseases (11/13) and cardiovascular disorders (11/17) are enriched in cluster-3. Note that diseases of other classes are not counted in the labels. Cluster details are in Table D in S1 Text.

### Inflammatory response subnetworks provide disease-specific insights

To gain a more comprehensive understanding of inflammatory components, we identified the subnetworks formed by the predicted cytokines and pathogenesis genes of a given disease. We inspected the subnetworks of five diseases representing immune disorders, infections, cardiovascular diseases, metabolic disorders, and neoplasms (systemic lupus erythematosus (SLE), TB, aneurysm, metabolic syndrome X, and acute leukemia) and found that they visually show different network patterns. SLE displays two heavily connected subnetworks centered on inflammatory cytokines and chemokines, while the network of TB shows that chemokines are distant from pathogenesis genes (Fig I in S1 Text). We quantified these network features by counting the number of high confidence associations between disease-associated genes (taken from DisGeNET) and cytokines in the five example diseases (Table 5). For these five diseases, 8–14% of the known disease-associated genes in DisGeNET are predicted to interact with at least one of the 126 cytokines, suggesting that the connections between pathogenesis and immune response are less dense compared to the connections between metabolic or signaling modules.

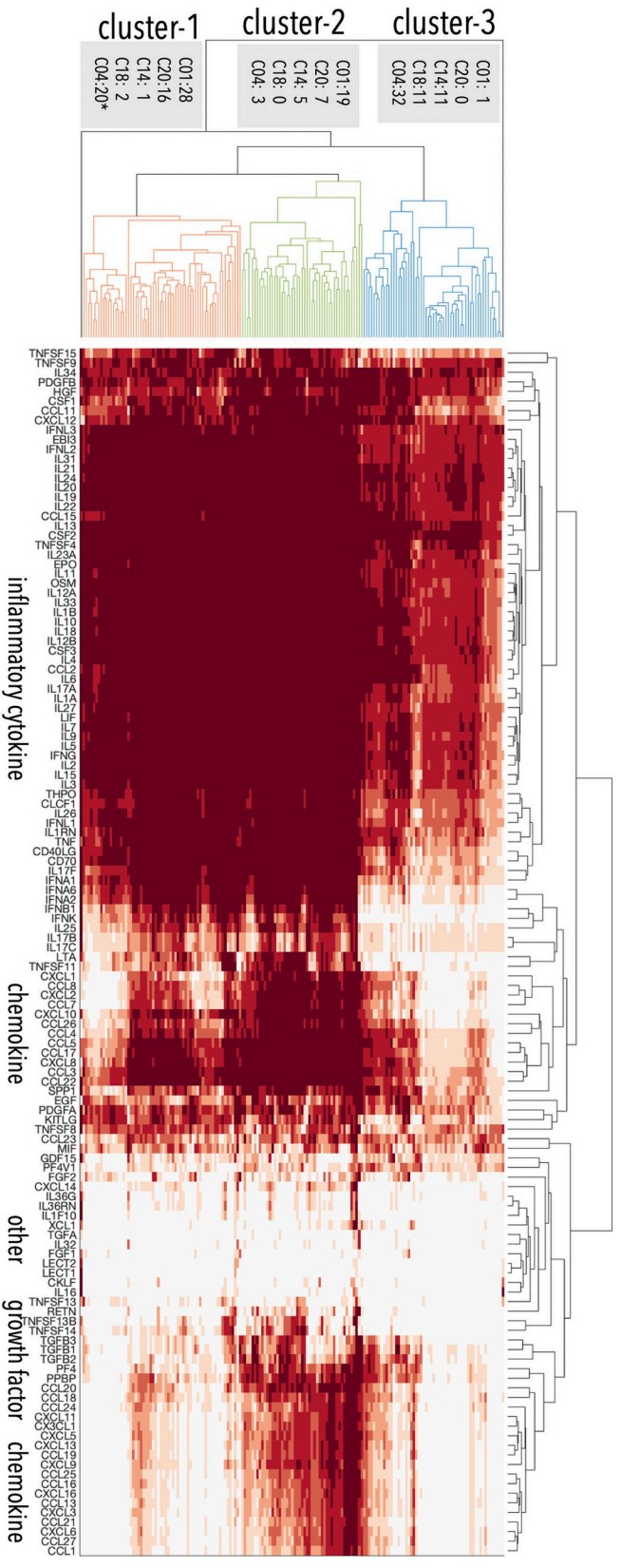

**Fig 8. Immune Scores of five disease classes (23 immune disorders, 48 infections, seventeen cardiovascular, thirteen metabolic, 55 neoplasms).** For each class, the average NAAS between its diseases and the cytokines within four categories are plotted: 47 inflammation related cytokines, 37 chemokines, thirteen growth factors, and 29 other cytokines. The chemokine scores for immune disorders are spread in a wide range. Growth factors have the highest scores in infections. Cardiovascular diseases have higher scores than metabolic diseases over the three groups of cytokines. Neoplasms show the lowest scores for all four categories.

The genes involved in the pathogenesis of a given disease often are cytokine receptors–and these are critical clues for how the pathogenesis genes may communicate with immune modules. The numbers of cytokine receptors that interact with cytokines can serve as a measurement for the density of inflammatory responses. For example, cytokine receptors of which their associations with diseases are known in DisGeNET are more heavily involved in the process from pathogenesis to inflammatory responses in SLE and TB, but not in aneurysm, metabolic syndrome X or acute leukemia. To capture the density of connections between pathogenesis genes and immune response, we compute an "Immune Connection Density (ICD)". We adapted the original equation of network efficiency [25] by counting the edges between the two components within the predicted subnetworks: genes for pathogenesis and genes for inflammation (see Methods). The interactions between cytokines, or between pathogenesis genes, are not included in this calculation. The ICD thus captures the associations between the two functions, pathogenesis and inflammation processes. Compared to SLE, aneurysm, metabolic syndrome X, and acute leukemia, TB shows the highest ICD (Table 5), suggesting that it triggers the most coherent reactions of immune systems upon infections.

Analyzing the subnetworks formed by the predicted cytokines and pathogenesis genes also suggested the reliability of the predicted disease profiles of 126 cytokines (See Result section "Cytokine features of diseases"). Some of the known genes from DisGeNET are cytokines. When comparing with all the cytokines known in DisGeNET, the recall rate (Table 6) of being recognized by the high confidence interactions ranges from 50% to 88%. When using lower confidence cutoff (0.7), 36 more essential cytokines are predicted to be associated with aneurysm, and 11/14 (78%) of the known essential cytokines are recognized by our network methods.

## Pathogenesis genes for immune disorders connect to key inflammatory genes

We observe different predicted cytokine associations across the five immune disorders in our dataset (Fig 9). The five diseases—rheumatoid arthritis (RA), psoriasis (PS), ulcerative colitis (UC), Crohn's disease (CD) and SLE—exhibit similar associations with the core inflammatory

**Table 5. Analysis of subnetworks formed by high confidence associations between the known disease associated genes and the predicted cytokines of five diseases.** The known DisGeNET genes (column #2) of a given disease often contain cytokine receptors. The number of cytokine receptors and other disease genes captured by high-confidence associations (column #3) is listed in column #4 and column #5, respectively. The number of predicted essential cytokines that interact with receptors and disease genes from DisGeNET is listed in column #6. The Immune Connection Density (ICD) estimated on the subnetworks formed by receptors, disease genes, and essential cytokines is shown in column #7.

| Disease | All DisGeNET genes | High confidence association | DisGeNET genes that interact with predicted essential cytokines | | | ICD |
|---|---|---|---|---|---|---|
| | | | Cytokine receptors | Disease genes | Predicted cytokines | |
| SLE | 793 | 1053 | 38 (5%) | 96 (12%) | 102 | 0.067 |
| TB | 135 | 213 | 9 (7%) | 12 (9%) | 70 | 0.123 |
| Aneurysm | 136 | 70 | 3 (2%) | 19 (14%) | 55 | 0.049 |
| Metabolic Syndrome X | 461 | 572 | 7 (2%) | 66 (14%) | 82 | 0.052 |
| Acute Leukemia | 425 | 307 | 11 (3%) | 35 (8%) | 90 | 0.062 |

**Table 6. The recall rate (column #4) of these cytokines being recognized by the predicted subnetworks formed by high-confidence associations ranges from 50% to 88%.**

| Disease | Known cytokines (DisGeNET) | Cytokines recognized | Recall rate |
|---|---|---|---|
| SLE | 50 | 44 | 88% |
| TB | 23 | 18 | 78% |
| Aneurysm | 14 | 7 | 50% |
| Metabolic Syndrome X | 17 | 13 | 76% |
| Acute Leukemia | 17 | 14 | 82% |

cytokines, but differ in their associations with chemokines, TNFs, and growth factors. For example, IL23A is highly involved (i.e., has a high NAAS) in all five diseases, while CCL18 and CCL7 are predicted to only associate with PS.

In order to understand how pathogenesis genes associate with different types of cytokines, we used a force-directed graph to visualize the various interactions in the process from pathogenesis to inflammatory responses under different disease contexts. Stronger associations are drawn with shorter "springs" in order to convey a qualitative understanding of the subnetwork structure [26]. The visualization in Fig 10 shows that the individual pathogenesis genes of SLE are closely linked to their embedded network neighbors—shorter edges correspond to closer network neighbors and thus allows us to assess the pathogenesis genes that are most likely associated with inflammation. Different sets (Box-C and Box-I-1) of pathogenesis genes form associations with chemokines and proinflammatory cytokines, respectively. Six genes (ACKR3, HRH4, HTR1, GAL, GRM3, S1PR1) are connected with a set of densely connected chemokines (23 cytokines marked in red box), with ANXA1 interacting with sixteen chemokines. Seven pathogenesis genes (Box-I-1) make interactions with a group of 28 proinflammation cytokines (orange box) directly or through receptors (green nodes). Next to this group of proinflammation cytokines, a group of nine disease associated genes (Box-I-3, S100A8, NLRP3, MYD88, IRAK1, IRAK4, TIRAP, TLR2, TLR5, TLR9) form a small clique with four

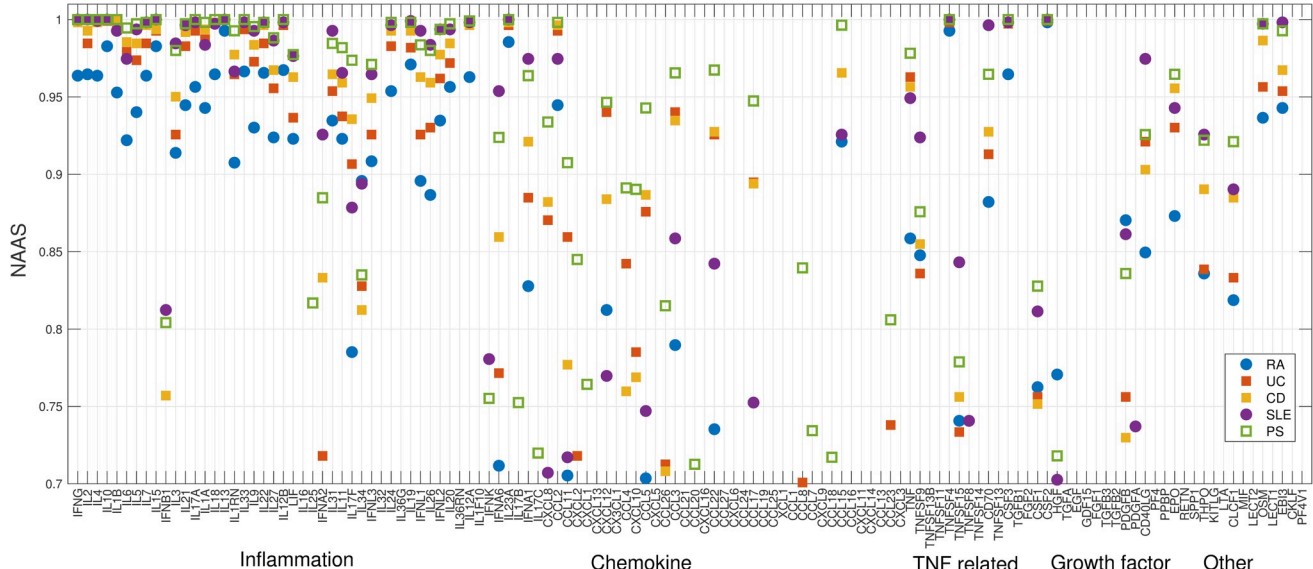

**Fig 9. Disease-specific cytokine profiles of five immune disorders.** The Y-axis shows the Probability of Association between each cytokine and the five immune disorders: rheumatoid arthritis (RA), psoriasis (PS), ulcerative colitis (UC), Crohn's disease (CD) and systemic lupus erythematosus (SLE). A conserved association pattern is observed in inflammation-related cytokines, while differential patterns are observed in other types of cytokines.

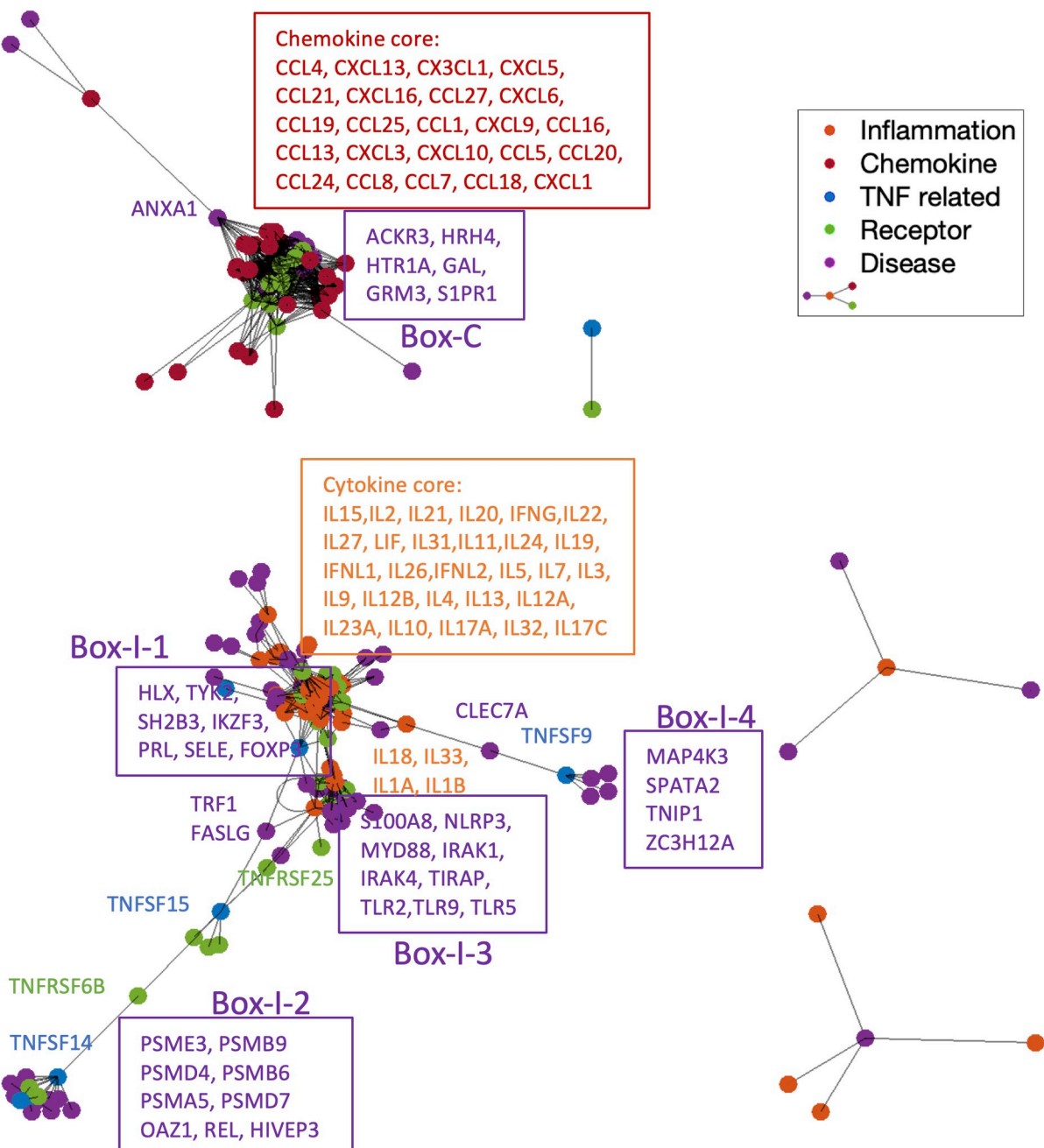

**Fig 10.** SLE subnetwork formed between pathogenesis genes (green and purple) and inflammatory responses (orange, red, blue). The graph was plotted using a force-directed layout that uses attractive forces between adjacent nodes and repulsive forces between distant nodes. The distances between two vertices are roughly proportional to the length of the shortest path between them. Six genes (ACKR3, HRH4, HTR1, GAL, GRM3, S1PR1) in Box-C are making high degree contacts with the chemokine core (red box), with ANXA1 interacting with 16 chemokines. Interactions with the inflammation core (orange box) appear in multiple directions. Seven pathogenesis genes (Box-I-1) interact with the inflammation core (orange box) directly or through receptors (green nodes). Nine disease genes in Box-I-3 form a small core with four cytokines (IL18, IL22, IL1A and IL1B). Two other groups of genes (Box-I-2 and Box-I-4) appear distant from the cytokine core but are linked to the TNFs, as they cannot overcome the repulsive force to association with the center of inflammation responses.

**Table 7. Pathogenesis genes in the highly connected modules that were identified by spectrum partition on the subnetworks formed by pathogenesis genes, receptors, and cytokines, in the context of five immune disorders: rheumatoid arthritis (RA), psoriasis (PS), ulcerative colitis (UC), Crohn's disease (CD) and systemic lupus erythematosus (SLE).** The table shows that 36 disease genes were identified out of 1,340 disease-associated genes for RA (2.7%), four disease genes from the 542 genes for PS (0.7%), 35 disease genes from the 793 genes for SLE (4.4%), eight disease genes from the 654 genes for UC (1.2%), and 18 disease genes from the 622 genes for CD (3%). Note that many of these disease-associated genes are related to immune responses.

| rheumatoid arthritis | | | psoriasis | systemic lupus erythematosus | | | | ulcerative colitis | Crohn's disease | |
|---|---|---|---|---|---|---|---|---|---|---|
| TRAF1 | TYK2 | NFKBIA | FASLG | TLR9 | TRAF1 | SH2B3 | | CFLAR | MAP3K1 | TYK2 |
| FASLG | OSMR | SIGIRR | PSMD7 | IRAK4 | FASLG | CR2 | | BIRC2 | TLR4 | GHR |
| MAP3K5 | GH1 | TSLP | REL | NFKBIA | TLR4 | SELE | | BIRC3 | TLR5 | GH1 |
| MALT1 | PRL | FOXP3 | LTBR | SIGIRR | TLR5 | CTLA4 | | NFKB2 | IRAK1 | PTPN2 |
| TLR4 | IKZF3 | TBX21 | | FOXP3 | IRAK1 | IRF9 | | USP14 | TLR1 | CTLA4 |
| TLR5 | BCL6 | CLEC7A | | TBX21 | S100A8 | IRF7 | | REL | IRAK3 | INPP5D |
| IRAK1 | SH2B3 | MAP4K3 | | CLEC7A | NLRP3 | HLX | | PSMG1 | NLRP3 | FOXP3 |
| S100A8 | SELE | PDCD5 | | MAP4K3 | TIRAP | IFIT1 | | EGLN3 | TLR2 | TSLP |
| TLR1 | CTLA4 | SPATA2 | | SPATA2 | MYD88 | IFI44 | | | TLR9 | NFKBIA |
| NLRP3 | CR2 | TNIP1 | | TNIP1 | TLR2 | IRF1 | | | | |
| TIRAP | PTPN2 | DDAH1 | | ZC3H12A | TYK2 | IRF2 | | | | |
| MYD88 | INPP5D | TLR2 | | OSMR | IKZF3 | IRF5 | | | | |
| | IFI4 | TLR9 | | PRL | | | | | | |

other proinflammation cytokines (IL18, IL22, IL1A and IL1B). Two other groups of genes (Box-I-2: PSME3, PSMB9, PSMD4, PSMB6, PSMA5, PSMD7, OAZ1, REL, HIVEP3, and Box-I-4: MAP4K3, SPATA2, TNIP1, ZC3H12A) are distant from these proinflammation cytokines (orange dots), even they are apparently linked to the TNFs.

### Highly connected graphs capture connections between immune disorder pathogenesis and inflammation

The ICD for the five immune disorders are: RA, 0.055, UC, 0.069, CD, 0.071, SLE, 0.067, and PS, 0.060, suggesting that density of association between pathogenesis and inflammation is similar. However, different sets of genes are involved in these interactions. For each specific immune disorder, we are interested in the key pathogenesis genes that mediate cytokine connections. From the predicted inflammatory response subnetworks, spectrum partition enabled identification of highly connected graphs, or modules formed by a group of well-connected cytokines and pathogenesis genes, revealing the key genes for cytokine mediators that drive the pathogenesis in inflammatory diseases. Table 7 shows that 23 receptors and 36 disease genes (from 1340 genes taken from DisGeNET) were identified to form well-connected cytokine modules in RA, six receptors and four disease genes (from 542 genes) for PS, seventeen receptors and 35 disease genes (from 793 genes) for SLE, four receptors and eight disease genes (from 654 genes) for UC, and thirteen receptors and eighteen disease genes (from 622 genes) for CD. This process further prioritizes the known disease associated genes from DisGeNET, providing a more focused set of candidates for experimental follow-up. Of additional note, the cytokine modules identified for SLE and RA overlap, while those identified for PS and UC overlap, but not the corresponding pathogenesis genes (Tables C-F in S1 Text), suggesting different mechanisms mediating the cytokine framework.

## Discussion

### A complete, large-scale map of associations between genes enables the identification of genome-wide features of cytokine interactions

The connection density between immune response genes is lower than that within the heavily studied modules for metabolic, transcriptome, and signaling. Meanwhile, the associations across functional units are not well defined, relative to the connections within known

functional units (Table A in S1 Text). Disease associated genes are often found scattered across different modules (metabolic, signaling, and immune modules). For example, for non-alcoholic steatohepatitis, disease associated genes are found in the highly disparate functional modules of fatty acid beta-oxidation, proteolysis, signal transduction, leukocyte aggregation, and other cellular process [17]. In this work, we aimed to identify novel associations between pathogenesis genes and immune responses; for this task, we required a map of pairwise associations between genes at a large scale. The STRING network itself is a highly connected graph that obeys "small world" statistics and thus path length calculations are not useful for estimating likelihood of association [21]. We cannot distinguish pairwise importance by shortest path length because there are too many gene pairs that share the same length. Our proposed embedding space provides more information by capturing the topological structures of STRING, thereby enabling a complete map of pairwise associations.

The high degree of pleiotropy and redundancy among cytokine family (each cytokine has multiple functions, and each function potentially mediated by multiple cytokines) make the classification of cytokines a challenge [23,27]. Using our map of pairwise associations, we were able to connect a key set of cytokines to 14,707 human genes and identified genome-wide features that interact with unique groups of cytokines. These genome-wide associations enable a more systematic classification of cytokines. The biological annotations of these specific interactions also provide important insights into the functions of cytokine groups. We identified two sets of genes (191 genes in SIG1 and 175 genes in SIG5) that interact only with chemokines, highlighting specific signaling pathways for chemokines: their biological functions focus on the G-protein signaling pathways, a response to endogenous and environmental insults. We also found that genes responsible for ubiquitin proteasome pathways interact with TNFs, not other cytokines (SIG3 in Table 1). SIG4 is another interesting gene set which interacts only with TGF and plays an important role in blood coagulation and plasminogen activating cascades, which are often associated with innate immunity in infectious and neuroinflammatory diseases. Some of the genes in the featured interactions (F5 and SERPINE2 in SIG4) are known to affect the concentrations of circulating cytokines [28]. Those genes that are not recognized by GWAS could be critical links from pathogenesis to inflammation. The biochemical pathways underlying the links from these genes to complex diseases have remained elusive. Our findings provide candidate genes pivoting to deeper studies of pathogenesis and inflammation.

## Disease-specific cytokine profiles reveal flexible features of differential inflammatory responses

Our analysis suggests that diseases have flexible cytokine distributions even though they may share cytokine framework that provides conserved mechanisms of inflammation.

First, clustering diseases based on their cytokine profiles yields three different cytokine response modes which correlate with disease classification. Of the 55 neoplasms studied, 32 fall into cluster-1 and twenty in cluster-3, of which nineteen are hematologic and lymphatic diseases (C15/C04) (Fig 7). Second, Immune Scores that capture the contributions from different types of cytokines to the inflammation show that inflammation is a driver of pathology for many diseases beyond those that are typically considered autoimmune or infectious. Cardiovascular diseases show higher Immune Scores than metabolic disorders and neoplasms (Fig 8). The increased concentrations of cytokines in cardiovascular diseases are not only markers of chronic low-grade inflammation, but also provide an important pathophysiological link between cardiovascular health and ageing [29]. Third, the number of disease genes and receptors that are associated with essential cytokines varies widely compared with the numbers of

cytokines themselves, suggesting that different mechanisms mediate between pathogenesis and inflammatory response (Table 5). Finally, ICDs which quantify the density of interactions between pathogenesis and inflammation suggest the mechanism by which different diseases have different levels of inflammation (Table 5).

Within the class of immune disorders, we also observed differential cytokine distributions between different diseases (Fig 9). The five immune disorders examined all show close interactions with the cytokines responsible for proinflammatory responses, but not all five of them have close interactions with chemokines, TNFs or growth factors. One explanation is that multiple cytokines are triggered simultaneously by a few key activated triggers. Therefore, identification of the key genes and cytokines that trigger the immune responses in individual diseases may provide insights into therapeutic strategies.

## Subnetworks between pathogenesis and inflammation suggest different mechanisms of immune response

Our predicted associations between cytokines and pathogenesis enable network analysis from different perspectives, providing useful insights into the molecular pathways that mediate inflammation. We investigated two methods for visualizing the subnetworks formed between pathogenesis and inflammation. Through hierarchical layered analysis on the connections between pathogenesis and inflammation, we were able to identify the central nodes in this dynamic process [30] (Fig J in S1 Text). For example, in the subnetwork for metabolic syndrome X, pathogenesis genes AGTR2 and ADRA1A are at the top hierarchical layer for chemokine signaling, while IRF1, MXZB1, MTTP, and CNTC are at the top layer in cytokine signaling. Additionally, our layered graph analysis suggests different interaction patterning: aneurysm showed a clear hierarchical flow starting from disease genes to cytokines, while metabolic syndrome X showed interactive layers between disease genes and cytokines, with an emphasis on chemokine responses, suggesting different mechanisms in signaling between pathogenesis and inflammation. We also utilized a force-directed graph to present the various interactions under different disease contexts [26] (Fig I in S1 Text). The networks for SLE and TB display different patterns, suggesting different mechanisms in triggering inflammatory responses in immune disorders and infections. These mechanisms may be related to the speed or strength of the immune reactions. Attractive forces between pathogenesis and chemokine responses are prominent in metabolic syndrome X, but not in aneurysm or acute leukemia. Interestingly, recent research has found that modification in the genes that closely interact with chemokines may affect functions in glucose and lipid metabolism in patients with metabolic syndrome X [31,32]. Our subnetwork for metabolic syndrome X provides candidates as novel targets for broader and more efficacious treatments and prevention of metabolic disease.

## Spectrum partition of subnetworks identifies key mediators of immune disorders

Immune cells can release many pathogenic cytokines. Mechanistic studies will be necessary to identify the key cytokines for a given inflammatory disorder and to pinpoint which cytokines might be the appropriate targets for tacking each disease. Given a disease, our methods identify the well-connected subnetwork formed between pathogenesis and inflammation and can extract key genes closely associated with cytokines within the subnetwork. These key genes can then serve as therapeutic target candidates, as they are predicted to be the main mediators of inflammation. Human trials targeting different cytokines have shown differential efficacy of cytokine inhibition in chronic inflammatory diseases. Most of the chronic inflammatory diseases share clinical responsiveness to TNF-a inhibition but differ in their responsiveness to

inhibition of cytokines, such as IL6, IL1, IL17 and IL23. This suggests the existence of a hierarchical framework of cytokines that defines groups for chronic inflammatory diseases, in contrast to the previously assumed the homogenous molecular disease patterns [15]. Interestingly, we have identified a common well-connected subnetwork that defines the close interactions between pathogenesis genes and cytokines in SLE and RA, which comprises pathogenesis genes TNIP1, SPATA2, MAP4K3, and CLEC7A. Annotation of these genes explains the possible shared pathways in SLE and RA, therefore shared therapeutic targets. Among these genes, TNIP1 is involved in inhibition of nuclear factor-κB (NF-κB) activation by interacting with TNF-α induced protein 3, an established susceptibility gene to SLE and RA [33]. Other evidence suggests that the downregulation of SPATA2 augments transcriptional activation of NF-κB and inhibits TNF-α-induced necroptosis, pointing to an important function of SPATA2 in modulating the outcomes of TNF-α signaling, which plays important roles in inflammatory responses in RA and SLE [34]. These observations support our predicted key mediators for pathogenesis and inflammation. Further study of other key genes identified from disease-specific subnetworks may provide additional insights into therapeutic strategies.

Our methods identify networks of cytokines and disease-related genes specific to each inflammatory disease. We cannot determine if these are the causal factors for disease specific clinical phenotypes without further analysis, including experimental investigations. However, our predictions provide insights into the potential underlying molecular mechanisms, and may be useful to guide experimental programs. In addition, tissue-specific effects are lost in using a unified PPI network. Our future work should focus on using tissue-specific PPI networks to refine our predictions as more comprehensive tissue-specific networks are made available, our future work should focus on using tissue-specific PPI networks to refine our predictions.

## Methods

### Network embedding of 14K human genes

We downloaded the network of 19,344 human genes in the STRING database. This network contains 5,879,727 total edges. We selected 14,707 genes that involve 728,090 high confidence edges (at a cutoff of 800) in STRING (Table A in S1 Text). We applied the methods of network scalable feature learning [22] to capture network topology features of the 14,707 genes in a 64-dimensional embedding space. Specifically, we have conducted a grid search over hyperparameters to identify the optimal settings for the embedding algorithm (Method Notes in S1 Text). The finalized parameters were as follows: for each node, we used it as source to sample ten paths, with each path at a length of thirty (We set the hyperparameters as length of walks = 30, number of walks = 10, min count = 1, batch word = 6, window = 10.). We then applied node2vec to this data to get a 64-dimensional embedding representation for each node.

### Prediction of association scores

For every pair of genes in our pool of 14,707 human genes, we calculated the cosine similarity between the two genes' embedding vectors. We refer to the resulting 108,140,571 pairwise scores as Association Scores. The STRING confidence score for 9,250,034 of these pairs was available, of which 8,521,944 pairs had confidence scores below 800 and thus were not used for embedding. We evaluated the predictive strength of Association Scores by comparing them to the known confidence scores for these 8,521,944 pairs.

### Prediction of disease-specific cytokine profiles

We identified 126 "essential cytokines" by mapping cytokines from ImmuneXpresso with the 14,707 human genes. For each cytokine gene, we calculated its Association Scores with each of the other 14,581 non-cytokine human genes. We collected 11,944 disease concepts from DisGeNET [24] that were associated with at least two genes in our set of 14,707 genes. For each disease concept, we calculated the average Association Score between its associated genes and each of the 126 cytokines, resulting in a 126-dimensional vector of Association Scores for each disease. The 11,944 diseases were grouped into four bins based on the number of genes associated with the disease: 2–9, 10–19, 20–49, or >49 (Fig F in S1 Text). The 126 Association Scores for each disease were normalized by the bin that the disease fell into: p-value = [number of diseases of which average cosine similarities <given distance] / [total number of diseases in the bin]. We refer to these scores as the disease's normalized average Association Scores (NAAS), or the disease-specific profile.

We collected 171 well-studied diseases of which the literature sampling frequency of 79 cytokines are available in ImmuneXpresso for validation. For each disease, we compared the predicted NAAS with the known literature sampling frequencies by calculating the Spearman correlation coefficients, where MatLab computes p-values for Spearman's rank correlation coefficient using the exact permutation distributions.

### Analysis of disease-specific cytokine profile features and subnetworks between pathogenesis and inflammation

Given a disease, we calculated the NAAS with respect to each of the 126 cytokines, resulting in a 126-dimensional disease-specific cytokine profile. We analyzed the features of these disease-specific cytokine profiles via hierarchical clustering. These disease-specific cytokine profiles were further converted into "Immune Scores" by averaging the NAAS for 47 inflammation related cytokines, 37 chemokines, 13 growth factors, and 29 other cytokines to quantify the contribution from four aspects of inflammatory responses.

In order to visualize the subnetwork formed between pathogenesis (disease associated genes) and inflammatory responses (essential cytokines), we labeled the disease genes as either "disease-specific" or "cytokine receptors" by mapping to the 110 cytokine receptors (available at https://github.com/TianyunC/cytokine-networks) that we defined by filtering genes acquired from GeneCards [35]. We graphed the subnetwork formed by high confidence connections (Association Score > 0.8) between disease genes and cytokines in a force-directed visualization that uses attractive forces between adjacent nodes and repulsive forces between distant nodes. The distance between two vertices in the graph is roughly proportional to the length of the shortest path between them within the subnetwork. To quantify the information exchange between pathogenesis and inflammatory responses, we calculated Immune Connection Density (ICD) with the formula $L = \frac{1}{N_p N_i} \sum_{pi} d_{pi}$, where $N_p$ is the total number of total pathogenesis genes in the disease-specific network, $N_i$ is the total number of total cytokines in the disease-specific network, and $d_{pi}$ is the cosine similarity of the two sets of genes in embedding space [25].

### Identification of highly connected graphs between pathogenesis and inflammation by spectrum partition

For a given disease, we constructed a graph G using the predicted subnetworks derived from the high confidence interactions between pathogenesis genes, receptors, and cytokines, with the interactions between cytokines, and those between pathogenesis genes removed. We then

calculated the Laplacian matrix L for the graph $G$, which yields a square, symmetric, sparse matrix. The smallest non-null eigenvalue of $L$ is called the Fiedler value, which represents the algebraic connectivity of a graph; the further from zero the Fiedler value, the more connected the graph. The Fiedler vector $w$ is the eigenvector corresponding to the smallest non-null eigenvalue of the graph. With this vector $w$, one can partition the graph into two or three sub-graphs using the Fiedler vector $w$. A node is assigned to one subgraph if it has a positive value in $w$ (well-connected nodes). Otherwise, the node is assigned to another subgraph (poorly con-nected nodes). Alternatively, the nodes of close to zero values in $w$ can be placed in a class of their own (known as articulation point). This practice is called a "sign cut" or "zero threshold cut". The sign cut minimizes the weight of the cut, subject to the upper and lower bounds on the weight of any nontrivial cut of the graph [36].

## Supporting information

**S1 Text. Supplementary material.** Method notes for embedding and grid searching. Fig A: Network sparsity within and between known functional modules in protein-protein interac-tion (PPI) networks. Fig B: The predicted Association Scores (Y-axis) of 8,521,944 edges corre-late with their known STRING confidence scores. Fig C: The distribution of Association Scores. Fig D: Predicted Association Scores correlate with known confidence scores in STRING. Fig E: The histogram of the number of genes associated with each of the 171 diseases. Fig F: The NAAS distribution within each bin defined by the number of genes associated with a disease. Fig G: The Number of genes associated with diseases is plotted against the P-value estimating the correlation between the predicted NAAS and the literature sampling frequency of cytokines. Fig H: The NAAS between aneurysm and each of the 79 cytokines are plot against the literature sampling frequency in aneurysm. Fig I: Graph plots showing interactions between pathogenesis genes and inflammatory responses. Fig J: Hierarchical structure showing information flow from pathogenesis genes to inflammatory responses. Table A: Statistics of selected sets from STRING and three classes of functional modules. Table B: Gene signatures identified for the six groups of cytokines. Table C: The predicted cytokine profiles correlate with the known literature sampling frequency in ImmuneXpresso for the 171 well-studied dis-eases. Table D: The 171 diseases classified into three clusters based on their cytokine profiles. Table E: Disease-associated genes in the well-connected modules formed by pathogenesis genes, receptors, and essential cytokines identified by spectrum partition in the context of five immune disorders. Table F: Frequency (#) in the five diseases.
(DOCX)

## Author Contributions

**Conceptualization:** Tianyun Liu, Russ B. Altman.

**Data curation:** Tianyun Liu, Shiyin Wang.

**Formal analysis:** Tianyun Liu.

**Funding acquisition:** Russ B. Altman.

**Investigation:** Tianyun Liu, Russ B. Altman.

**Methodology:** Tianyun Liu.

**Project administration:** Tianyun Liu, Russ B. Altman.

**Supervision:** Russ B. Altman.

**Validation:** Tianyun Liu.

**Visualization:** Tianyun Liu.

**Writing – original draft:** Tianyun Liu.

**Writing – review & editing:** Michael Wornow, Russ B. Altman.

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
