## [Decision Letter · Decision Letter 0]

12 Nov 2021

Dear Dr. Altman,

Thank you very much for submitting your manuscript "Construction of disease-specific cytokine profiles by associating disease genes with immune responses" for consideration at PLOS Computational Biology.

As with all papers reviewed by the journal, your manuscript was reviewed by members of the editorial board and by several independent reviewers. In light of the reviews (below this email), we would like to invite the resubmission of a significantly-revised version that takes into account the reviewers' comments. Please make sure to address all concerns sufficiently including the data and code availability, use of proper statistical methodology, revising the support for each major claim and the consistency of terminology in the text.

We cannot make any decision about publication until we have seen the revised manuscript and your response to the reviewers' comments. Your revised manuscript is also likely to be sent to reviewers for further evaluation.

Sincerely,

Ferhat Ay, Ph.D

Associate Editor

PLOS Computational Biology

Rob De Boer

Deputy Editor

PLOS Computational Biology

Reviewer's Responses to Questions

**Comments to the Authors:**

Reviewer #1: Here the authors ask "how genes involved in pathogenesis which are often not associated with the immune system in an obvious way communicate with the immune system?". They further ask "What is the degree to which cytokine

responses are shared across diseases or specific to each disease? And how does heterogeneity of

cytokine responses mediate different pathogenesis and inflammatory processes?". These are important mechanistic questions. Their comprehensive analysis reveals subnetworks formed between disease-specific pathogenesis genes, hormones, receptors, and cytokines, leading to genes responsible for interactions between pathogenesis and inflammatory responses. In line with their observations, trials indicated a hierarchical framework of cytokines that defines groups for chronic inflammatory diseases rather differently from the homogenous molecular disease pattern previously assumed. The authors detail and annotate the genes that they identified and their functions. Altogether, this is a good, coherent paper that is well-suited to PLOS CB. Especially, I like it since it aims at elucidating mechanisms.

It is always possible to raise technical, methodological and/or statistical questions. In this case I do not think that these are needed. The manuscript describes an in-depth, thoughtful, robust and innovative approach and its output, and I hope that the results will indeed be useful toward drug discovery.

Reviewer #2: The manuscript entitled “Construction of disease-specific cytokine profiles by associating disease genes with immune responses” by Liu et al submitted to PLoS Computational Biology for publication (Manuscript Number: PCOMPBIOL-D-21-01684) presents an interesting attempt at discovering pathways of interaction between disease pathogenesis genes and immune-response driving cytokines and shed light on disease-specific mechanisms that mediate pathogenic immune-responses using an innovative network-based approach. However, there are a number of issues that need to be addressed before considering the manuscript for publication.

Minor comments:

Introduction:

- Page 3, lines 1-3: please provide references to support the claim made in the sentence.

- Paragraph 5: please move the sentences describing the results presented in Supplementary Material Figure S1 either to the results section or to the method sections, where they can be used to justify the usage of the network embedding procedure rather than the STRING PPI-network itself.

Results:

- Please represent the results presented in Figure 2A as a “STRING confidence score vs association score” density map scatterplot, so the reader can have a clearer idea of how those two metrics track each other. Also, please calculate a correlation coefficient and add it to the results.

- Please perform a multiple-testing correction on the p-values obtained in the NAAS vs ImmuneXpresso literature co-occurrence frequency correlation analysis (Figure 4A) to decide which of the 171 disease show a statistically significant correlation.

- Please represent the results presented in Figure 4B either as a Normalized Average Association Score vs literature co-occurrence frequency (in log scale?) scatterplot or, alternatively, as boxplot showing the distribution of NAAS for those disease-cytokine pairs with literature co-occurrence frequency above 0.005 and below 0.005 separately.

- When presenting the results of the overlap between the high NAAS and high literature co-occurrence frequency, please add Specificity and overall Accuracy besides the recall.

- Subsection “Defining the key modes of cytokine response”, paragraph 1:

o The authors state that “They fall into three distinct patterns in cytokine response”. According to what? The results of a hierarchical clustering? I know the readers can infer that from looking at Figure 5A but, by just reading the text, the assertion seems arbitrary. Please add detail.

o The description of the results of Figure 5 in the main text and in the figure’s legend do not agree. Please revise so both pieces of text describe the results in the figure appropriately.

- Subsection “Defining the key modes of cytokine response”, paragraph 2:

o It’s unclear what is meant by the first sentence. Please rephrase to make it clearer.

o Please use ‘drive’ instead of decide in the third sentence of this paragraph.

o In Figure 5A, it is not possible to identify the different groups of inflammatory cytokines (interleukins, interferons and TNFs). Please use color-coding on the cytokine labels to aid the identification of those subgroups of cytokines.

o Please try and summarize the results presented in figure 5B more concisely. In general, immune disorder and infections, as expected, show higher values for all 4 cytokine-type components, with cardiovascular diseases presenting intermediate values, and metabolic diseases and, specially, neoplams, having the lowest average immune scores for all 4 components.

- Subsection “Inflammatory response subnetworks provide disease-specific insight”, paragraph 1:

o The fact that the subnetworks identified for each disease look different on visual inspection is a meaningful result. Please either remove that part of the sentences or elaborate in more detail into the differences between the disease subnetworks.

o Please explain the rationale for the assertion made in the last sentence of the paragraph. Would you expect to see more connections if, instead of cytokines, the interactors were metabolic or signaling genes? And, if so, what would you base your expectation on?

- Subsection “Inflammatory response subnetworks provide disease-specific insight”, paragraph 2:

o Please clarify if, for any given disease, only the cytokine receptors that are associated with the disease as per DisGeNET or, alternatively, all cytokine receptors that appear in your embedded network are included in these subnetwork.

o “Therefore, the ICD suggests the associations between…” please use some other term instead of “suggests”: “captures” or “represents” could do the job, for example.

o It would be informative to assign p-values to the ICD values you calculate for each disease (Table 2A). For that, please, perform simulations, for each disease, by taking, randomly, as many non-cytokine genes in the network as the number of the DisGeNET genes for the disease that appear in your global network, calculate the ICD for the resulting subnetwork and, after repeating this procedure X times (at least 1000, preferably), calculate the percentile for your real ICD value to get a p-value. Repeating this procedure for each of the five diseases should assign a p-value for each ICD and aid its interpretation.

- Subsection “Immune disorders pathogenesis genes connect to key inflammatory genes”, paragraph 1:

o In Figure 6, please explain in the legend that only NAAS values > 0.7 are shown. Also, please use 2-3 cytokines as examples to elaborate on and illustrate the fact that all 5 disease show similar patterns of association with inflammatory cytokines, but very varying levels with chemokins, TNFs and growth factors (Il18 and CCL17 look like they could be good candidates for that!).

- Subsection “Immune disorders pathogenesis genes connect to key inflammatory genes”, paragraph 2:

o Page 10, line 1: there is probably a typo and “Box-I-1-4” should probably read “Box-I-1”. Please check and fix if necessary.

o From this point on, there are several instances where pre-inflammatory cytokines are mentioned. I assume this is a mistake and “pro-inflammatory” cytokines is the appropriate term. Please fix all instances if this is the case.

- Subsection “Highly connected graphs capture connections between immune disorder pathogenesis and inflammation”, paragraph 1:

o Is there a typo in the first sentence of is CD’s ICD really 0.71? Please check and amend if necessary.

o In supplementary table S5, it's impossible to infer from the gene-lists how many are shared across the diseases. Please add either venn diagrams showing the overlap between the gene-lists, or a list of unique geneIDs/symbols, indicating, for each gene, in how many diseases it appears and which are those diseases.

Discussion:

- Subsection “A complete map of associations between genes in large scale enables the identification of genome-wide features of cytokine interactions”, paragraph 1:

o It is difficult to understand the results presented in Supplementary Materials Table S1 without further explanation. For example, what do ME and HM stand for? How where those gene-sets obtained? Please add detail to aid understanding.

o From the first four sentences of the paragraph, it doesn't follow that one needs to identify and explore novel associations between disease pathogenesis genes and the immune response. Please add another sentence in between to link the context provided at the beginning of the paragraph and the stated goal of the study.

o Please explain why the fact that a network has "small world" properties makes it unsuitable for estimation of likelihood of association between genes via path-length calculations. For non-graph experts, this is not immediately obvious.

- Subsection “Subnetworks between pathogenesis and inflammation suggest different mechanisms of immune response”:

o The results of Supplementary materials Figure S9 should be presented in the results section rather than here in the Discussion.

- Subsection “Spectrum partition of subnetworks identifies key mediators of immune disorders”:

o “These key genes serve as candidates of therapeutic targets, as they are the main mediators to fuel inflammation”: This is a very strong assertion that, I feel, is not justified by the results. It is true that the subnetworks have allowed the identification of genes involved in pathogenesis that might have a notorious role in driving the immune responses in those diseases, but more work is needed to confirm that this is, indeed, the case, and that they could be good candidates for therapeutic intervention. Please rephrase to tone down.

o “The human trials targeting different cytokines suggest the existence of a hierarchical framework of cytokines that defines groups for chronic inflammatory diseases rather differently from the homogenous molecular disease pattern previously assumed”: I find this sentence very difficult to understand. Please rephrase to increase clarity.

o “These observations validate our predicted key mediators for pathogenesis and inflammation”: I think “validate” is a clear overstatement here. Please tone down to “support” or to another similar term.

- Please add a paragraph discussing the limitations of the study and another one discussing potential future steps regarding how to overcome those limitations and/or strategies for validation of the therapeutic candidate genes identified.

Methods

- Please provide details on the experiment performed to find the optimal hyper-parameters for the embedding algorithm and how the final parameters were chosen.

- Did you calculate the cosine distance or the cosine similarity? I mean, if the resulting Association Scores are positively associated with the edge-confidences in STRING, I suppose that cosine similarities were calculated, rather than distances (1-cos_similarity). Otherwise, the paper doesn't make any sense. Please clarify.

- As far as I can tell, there are 140 cytokines in ImmuneXpresso. Did only 126 of those map onto the 14707 human gene network? Please clarify.

- Please add explanations on why it is necessary to perform the embedding of the STRING network onto 64-dimensional space rather than using the STRING network itself to calculate potential disease-gene – cytokine associations.

- Why was necessary to normalize the average association scores for each cytokine and disease for gene-set size? Did you observe an association between the number of genes associated to a disease and the average Association Scores between the genes and the cytokines? If so, please provide some result showing this association.

- It is not clear to me how the cytokine profiles are normalized by the associated-gene-count-bin each disease falls into. For starters, in each bin, is the calculation of the p-value done using the distribution of the association score across all cytokines or for each cytokine separately? Also, how is the normalized average Association Score calculated, then? Is it the -log10pvalue? Or the percentile? Please add details to clarify.

- For the validation of the NAAS scores, did you calculate the Spearman correlation between the NAAS profile across 79 cytokines for a disease and the corresponding profile of the literature sampling frequency for the disease? Please add clarification. Also, please explain the rationale behind using ImmuneXpresso's literature sampling frequency profile to validate the Association Score and an explanation of what the “literature sampling frequencies” represent. Are these frequencies of co-occurrence of a disease and a cytokine across the literature?

- How was the filtering done to reach the list of 110 cytokine receptors from GeneCards? Please provide details.

- “and is the cosine distance of two sets of genes in embedding space”. Two comment here:

o Again, please clarify if you calculated cosine distances or cosine similarities, as the analysis wouldn’t make sense as it stands if using the former.

o I would expect the dpi to represent the cosine distance (or similarity) in embedding space between each pair of pathogenesis genes and cytokines and not the overall cosine distance (or similarity) between the two sets of genes, right? Please clarify.

- If the object of the spectrum partition was to identify the pathogenesis genes showing a high connectivity with cytokines in each disease-specific subnetwork, I would expect one would have to remove all the interactions among pathogenesis genes, too, and not only the interactions among cytokines as, otherwise, the Fiedler vectors will reflect not only pathogenic gene – cytokine connectivity but, also, pathogenic gene – pathogenic gene connectivity. However, the text only makes reference to interactions between cytokines being removed. Please clarify this point.

Figures:

- In Figure 1A, please remove the arrow from the “14,707 genes” box to the “9,250,034 pairs in STRING” box as it suggests that the STRING interactions were identified based on the initial selection of those genes when, according to the explanation in the methods section, that is not the case.

Overall comments:

- Throughout the text inflammation/inflammatory and immune/immunological seem to be used interchangeably. My understanding is that the scope of the study is the exploration of the broader immune response rather than of the particular inflammatory aspect of immune responses. If that’s the case, please change all instances where inflammation/inflammatory is used to refer to the broader immune response. Otherwise, please clarify in the text that the study’s focus is inflammation only.

- There are a few instances in the text where the study is framed as identifying disease-specific cytokine response profiles. I find this misleading. No levels of cytokines or of their downstream effectors were used in the study. Please rephrase those instances to more accurately reflect the fact that the cytokine profiles represent network-interaction-based potential cytokine involvement rather than cytokine response.

- There are quite a few grammatical errors and some of the sentences are not worded in an easily understandable manner. Please get the manuscript proofread by a native English speaker to fix grammatical and punctuation mistakes and make some sentences clearer.

In this work, the concept of seeking to understand how genes involved in pathogenesis which are often not associated with the immune system in an obvious way communicate with the immune system is an endeavor of interest to the research community. Overall, the writing can be further refined for more concise and specific articulation and flow of concepts.

It is a useful endeavor to establish a threshold of high confidence interactions. Even in the case where the interaction can occur: ligand/receptor, it is not an indication that in a given disease tissue it does occur unless covariance and level of expression signifies there is a high probability or confidence level of this occurring in which case rna and protein expression would have to be used as supporting data.

• Another question to ask is how does the hierarchy of cytokine response differ or is conserved across diseases and can you characterize the various function all gene sets that are associated with specific cytokine responses.This was not done.

• It remains unclear whether inflammation score is differentiating/ qualifying the type of inflammatory response or whether it is quantifying the magnitude of immune response.

• Unclear what the rationale is for merging all diseases in the analysis between protein protein interaction and disease genes and then separating the diseases for inflammation score. The former may be dependent on the disease with some genes interacting in some diseases and not others whereas the magnitude of inflammation score may converge across diseases since it tracks with severity of inflammation. Gene interactions may also depend on tissue context which is associated with disease type. It is these context dependent interactions which may be the drivers of cytokine responses, with some collections of cytokines triggering dependencies on other cytokines so it maybe overly simplistic to start with one universal disease gene PPI framework. In this way, the ingoing assumption is flawed.

• The ‘known’ disease cytokine associations derived from NLP are questionable.

• Provide a build in of an in silico negative control.

• It would be interesting if this model could parse the relationships between what disease genes are causal to driving a cytokines, how the inter relationship of different cytokines are altered in different disease settings and then what is the disease gene effect of cytokine stimulation

• Some of the preliminary conclusions have not been vetted properly to identify biological meaning. For example ” Chemokine-CLU is associated with a set of genes that function in G-protein signaling pathway and the response to endogenous and environmental insults ". This is the case because chemokine receptors are GPCRs.

• “Note that of the twenty neoplasms in cluster-3, nineteen are hematic and lymphatic diseases (C15/C04), suggesting that these neoplasms have distinct cytokine distributions from other neoplasms. This is expected given hematopoietic and lymphatic are immune organs as compared to neoplasms of other tissues.

• “We also found that genes responsible for Ubiquitin proteasome pathways interact with TNFs, not other cytokines” already known in part and not entirely true.

• Main conclusions are either too vague or already known or recite existing open questions which have not been sufficiently investigated and analyzed in this model.

• Neoplasms (systemic lupus erythematosus (SLE), TB, aneurysm, metabolic syndrome X, and acute leukemia). These examples do not fit the definition of neoplasm.

• While the collection of methods are interesting, the novel conclusions are limited, i.e it is already known TLR activation is linked to cytokine/TNF response.

• Since RA, SLE and IBD are inflammatory diseases it is expected that they would have a higher ICD score than the other disease genes. The final conclusions between these diseases is vague and not analyzed with sufficient depth.

Reviewer #3: Manuscript PCOMPBIOL-D-21-01684

The article is framed in the potential relationship of inflammation on human disease and how to uncover this relationship. The authors present a method to obtain disease-specific cytokines and their associated disease-specific genes. The method is based on prediction of protein-protein interactions based on network embeddings and previous knowledge on inflammatory genes and disease genes.

The manuscript would benefit on more clarity on the goals and description of the methodology. It has been hard to understand the rationale behind key aspects of the methodology and its implementation. Moreover, data and computational code is not available for evaluating the results and their reproducibility.

Said that, from what I could understand, the proposed method is based on some assumptions that deserve a more careful thought. One is that there are disease-genes and disease-specific cytokines, and by mapping these gene sets to PPI networks it is possible to uncover relationships between inflammation and disease. This is fine with the exception that there are a lot of immune related genes and cytokines already associated to diseases. Thus, there is no such disjoint gene sets, and this is related to the complexity of biology and conflicts with our intention to classify processes in clear-cut boxes. The other caveat is the assumption of a complete graph for the protein interaction network. In my understanding biological networks are not complete graphs.

More specific comments are provided below.

1. A non-negligible number of disease genes are actually cytokines. How your method accounts for this overlap in gene sets? In this context, the separation of disease-gene and cytokine-gene sets seems rather artificial. Please elaborate on this.

2. Intro, p. 4“Ideally, we would like to map known immune response genes more completely to human gene networks to better identify potential links to pathogenesis genes.” Do you mean inflammatory disease genes? There are many expressions like this in the article that are not clear enough to understand what the authors mean.

3. Why selecting STRING as a source of PPI interaction networks? There are other resources that are available and include the latest datasets for discovering PPIs, such as Intact. In addition, STRING includes different types of data and methods to derive the associations, the authors should specify which associations were included.

4. What is the rationale of the network sparsity analysis commented in the Intro? Why should we expect a complete graph in a biological network? This analysis should be better explained and justified.

5. Figure S1 refers to “modules”, but no details on how these modules are detected is provided. The same for the reference to “functional modules”. How immune response and disease modules are obtained, for instance? Please define the labels in Fig S1. Why the methodology for this analysis is not included in the methods section is not clear.

6. Does the Association Score consider experimental results for any given interaction?

7. Figures S2-S4 refer to “network distance”, do you mean “association score”? It is quite difficult to follow the figures and the text is there is no consistency in the naming of variables.

8. Please explain which diseases were selected from DisGeNET (the 11,944 disease concepts) and the rationale for the selection.

9. Figure S5 is hard to interpret without labels in the axis.

10. The code and data are provided under this link https://simtk.org/projects/cytokine

But it is not possible to access to the data or code without being a member of the project. Therefore, the datasets and the code are not available to ensure reproducibility.

**Have the authors made all data and (if applicable) computational code underlying the findings in their manuscript fully available?**

Reviewer #1: Yes

Reviewer #2: None

Reviewer #3: **No: **The code and data are provided under this link https://simtk.org/projects/cytokine

But it is not possible to access to the data or code without being a member of the project. I have registered to the platform but still do not have access to the project. Therefore the datasets and the code are not available to ensure reproducibility.

PLOS authors have the option to publish the peer review history of their article (what does this mean?). If published, this will include your full peer review and any attached files.

Reviewer #1: No

Reviewer #2: No

Reviewer #3: No
---

## [Decision Letter · Decision Letter 1]

3 Mar 2022

Dear Dr. Altman,

Thank you very much for submitting your manuscript "Construction of disease-specific cytokine profiles by associating disease genes with immune responses" for consideration at PLOS Computational Biology. As with all papers reviewed by the journal, your manuscript was reviewed by members of the editorial board and by several independent reviewers. The reviewers appreciated the attention to an important topic. Based on the reviews, we are likely to accept this manuscript for publication, providing that you modify the manuscript according to the review recommendations.

In addition to the remaining reviewer comments, you will need to make the source code publicly available before we can accept your work for publication. Please check the Sharing Software section of the journal policy: https://journals.plos.org/ploscompbiol/s/materials-software-and-code-sharing 

Sincerely,

Ferhat Ay, Ph.D

Associate Editor

PLOS Computational Biology

Rob De Boer

Deputy Editor

PLOS Computational Biology

[LINK]

Reviewer's Responses to Questions

**Comments to the Authors:**

Reviewer #2: Please address the remaining issues in your analysis:

The pathogenesis genes which you cite as interacting with cytokines (immune genes) are in fact almost all immune genes because there are inflammatory diseases (Table 3)

CVD may not be an infectious or autoimmune disease, but it is widely considered a chronic inflammatory disease.

Aneurysm or leukemia may affect blood or vessel lining cells but they are not considered as immune mediated per se.

Many chemokine/ receptors are GPCRs so may be circular in your enrichment analysis.

**Have the authors made all data and (if applicable) computational code underlying the findings in their manuscript fully available?**

Reviewer #2: Yes

PLOS authors have the option to publish the peer review history of their article (what does this mean?). If published, this will include your full peer review and any attached files.

Reviewer #2: No

Figure Files:

Data Requirements:

Reproducibility:

References:

---

## [Editor Report · Decision Letter 2]

17 Mar 2022

Dear Dr. Altman,

We are pleased to inform you that your manuscript 'Construction of disease-specific cytokine profiles by associating disease genes with immune responses' has been provisionally accepted for publication in PLOS Computational Biology.

Best regards,

Ferhat Ay, Ph.D

Associate Editor

PLOS Computational Biology

Rob De Boer

Deputy Editor

PLOS Computational Biology

---

## [Editor Report · Acceptance letter]

8 Apr 2022

PCOMPBIOL-D-21-01684R2 

Construction of disease-specific cytokine profiles by associating disease genes with immune responses

Dear Dr Altman,

I am pleased to inform you that your manuscript has been formally accepted for publication in PLOS Computational Biology. Your manuscript is now with our production department and you will be notified of the publication date in due course.

With kind regards,

Katalin Szabo
